# DIALOGUE: A Generative AI-Based Pre–Post Simulation Study to Enhance Diagnostic Communication in Medical Students Through Virtual Type 2 Diabetes Scenarios

**DOI:** 10.3390/ejihpe15080152

**Published:** 2025-08-07

**Authors:** Ricardo Xopan Suárez-García, Quetzal Chavez-Castañeda, Rodrigo Orrico-Pérez, Sebastián Valencia-Marin, Ari Evelyn Castañeda-Ramírez, Efrén Quiñones-Lara, Claudio Adrián Ramos-Cortés, Areli Marlene Gaytán-Gómez, Jonathan Cortés-Rodríguez, Jazel Jarquín-Ramírez, Nallely Guadalupe Aguilar-Marchand, Graciela Valdés-Hernández, Tomás Eduardo Campos-Martínez, Alonso Vilches-Flores, Sonia Leon-Cabrera, Adolfo René Méndez-Cruz, Brenda Ofelia Jay-Jímenez, Héctor Iván Saldívar-Cerón

**Affiliations:** 1Unidad de Remisión de Diabetes Mellitus (URDM), Facultad de Estudios Superiores-Iztacala, Universidad Nacional Autónoma de México, Tlalnepantla 54090, Mexico; ricardoxopan@comunidad.unam.mx (R.X.S.-G.); 316191218@iztacala.unam.mx (S.V.-M.); ari_casra@comunidad.unam.mx (A.E.C.-R.); efrenql@comunidad.unam.mx (E.Q.-L.); cramosc2402@alumno.ipn.mx (C.A.R.-C.); 316152945@iztacala.unam.mx (A.M.G.-G.); 2Carrera de Médico Cirujano, Facultad de Estudios Superiores-Iztacala, Universidad Nacional Autónoma de México, Tlalnepantla 54090, Mexico; chavezcastanedaquetzal@unam.edu (Q.C.-C.); 423034277@iztacala.unam.mx (R.O.-P.); jazel.jarquin@iztacala.unam.mx (J.J.-R.); nallely.marchand@iztacala.unam.mx (N.G.A.-M.); graciela.valdes@iztacala.unam.mx (G.V.-H.); tecamposmtz@comunidad.unam.mx (T.E.C.-M.); vilches@unam.mx (A.V.-F.); soleonca@iztacala.unam.mx (S.L.-C.); armendez@unam.mx (A.R.M.-C.); 3Laboratorio de Medicina de la Conservación, Escuela Superior de Medicina, Instituto Politécnico Nacional (IPN), Plan de San Luis y Díaz Mirón, Colonia Casco de Santo Tomás, Miguel Hidalgo, México City 11350, Mexico; 4Colegio de Ciencias y Humanidades, Plantel (I) Azcapotzalco (CCH), Av. Aquiles Serdan No. 2060, Ex-Hcienda del Rosario, Azcapotzalco, México City 02020, Mexico; jonathancortes3241@alumno.cch.unam.mx; 5Centro Internacional de Simulación y Entrenamiento en Soporte Vital Iztacala (CISESVI), Facultad de Estudios Superiores-Iztacala, Universidad Nacional Autónoma de México, Tlalnepantla 54090, Mexico; 6Academia del Módulo de Sistema Endocrino, Carrera de Médico Cirujano, Facultad de Estudios Superiores Iztacala, Universidad Nacional Autónoma de México (UNAM), Tlalnepantla 54090, Mexico; 7Unidad de Biomedicina (UBIMED), Facultad de Estudios Superiores Iztacala, Universidad Nacional Autónoma de México, Tlalnepantla 54090, Mexico; 8Laboratorio de Inmunología (UMF), Facultad de Estudios Superiores Iztacala, Universidad Nacional Autónoma de México, Los Barrios N° 1, Los Reyes Iztacala, Tlalnepantla 54090, Mexico

**Keywords:** generative AI, medical education, diagnostic communication, ChatGPT, virtual patient, standardized patient, communication training, empathy in diagnosis, formative simulation

## Abstract

DIALOGUE (DIagnostic AI Learning through Objective Guided User Experience) is a generative artificial intelligence (GenAI)-based training program designed to enhance diagnostic communication skills in medical students. In this single-arm pre–post study, we evaluated whether DIALOGUE could improve students’ ability to disclose a type 2 diabetes mellitus (T2DM) diagnosis with clarity, structure, and empathy. Thirty clinical-phase students completed two pre-test virtual encounters with an AI-simulated patient (ChatGPT, GPT-4o), scored by blinded raters using an eight-domain rubric. Participants then engaged in ten asynchronous GenAI scenarios with automated natural-language feedback. Seven days later, they completed two post-test consultations with human standardized patients, again evaluated with the same rubric. Mean total performance increased by 36.7 points (95% CI: 31.4–42.1; *p* < 0.001), and the proportion of high-performing students rose from 0% to 70%. Gains were significant across all domains, most notably in opening the encounter, closure, and diabetes specific explanation. Multiple regression showed that lower baseline empathy (β = −0.41, *p* = 0.005) and higher digital self-efficacy (β = 0.35, *p* = 0.016) independently predicted greater improvement; gender had only a marginal effect. Cluster analysis revealed three learner profiles, with the highest-gain group characterized by low empathy and high digital self-efficacy. Inter-rater reliability was excellent (ICC ≈ 0.90). These findings provide empirical evidence that GenAI-mediated training can meaningfully enhance diagnostic communication and may serve as a scalable, individualized adjunct to conventional medical education.

## 1. Introduction

DIALOGUE (DIagnostic AI Learning through Objective Guided User Experience) is an educational intervention designed to train medical students in diagnostic communication through scalable, AI-based simulations. As artificial intelligence (AI) rapidly transforms medical education, generative AI models such as ChatGPT have gained attention for their ability to simulate human-like dialogue and provide individualized feedback ([7]). Unlike traditional digital tools, large language models (LLMs) can dynamically engage students in realistic, interactive patient encounters, potentially accelerating learning across clinical domains ([3]). A key advantage of these tools lies in their capacity to tailor educational experiences—analyzing individual learner profiles and adapting feedback to specific gaps and learning styles ([3]). Beyond knowledge delivery, generative AI enables the simulation of nuanced diagnostic conversations, fostering students’ critical thinking and communication skills in a cost-effective, high-frequency format. In short, these technologies offer new possibilities for scalable, on-demand clinical training in medical education ([3]). In short, generative AI has opened exciting possibilities for scalable, on-demand clinical training in medical education.

Communication skills, especially those related to diagnosis, are a core competency for physicians and a critical focus for medical training. Effective doctor–patient communication improves the quality of care, whereas poor communication can undermine history-taking, hinder accurate diagnosis, and negatively affect patient adherence to treatments ([14]). Importantly, deficiencies in communication—rather than medical knowledge—are often cited as a leading cause of patient dissatisfaction and complaints ([14]). Despite the well-recognized importance of communication, many trainees still feel underprepared in this domain. For example, a recent survey found that while over 70% of senior medical students had received some instruction in “breaking bad news,” only about 17% felt adequately prepared to actually deliver serious diagnoses to patients ([19]). Nearly all students in that study agreed on the necessity of being well prepared to communicate difficult news ([19]), highlighting a gap between current training and learner needs. Clearly, there is an educational imperative to strengthen medical students’ diagnostic communication skills—from effective history-taking and explanation of clinical reasoning to empathetic disclosure of diagnoses—through improved teaching methods and practice opportunities.

Simulation-based education has long been an integral strategy for teaching clinical communication and diagnostic reasoning skills in a safe, controlled environment ([6]). Traditional simulations often involve standardized patients (actors trained to portray patients), which have been shown to significantly improve learners’ diagnostic accuracy, communication abilities, and overall clinical competence ([6]). Simulated patient encounters have long been used to develop communication skills in medical training, particularly through objective structured clinical examinations (OSCEs). These face-to-face simulations are effective but resource-intensive, requiring extensive time, personnel, and coordination, which inherently limits their frequency and scalability ([22]). Previous attempts to digitize these interactions—using scripted virtual patients or rule-based chatbots—have yielded mixed results, often hindered by limited interactivity, lack of realism, and poor accessibility ([18]). The recent emergence of generative AI models capable of producing coherent, unscripted dialogue opens new possibilities for scalable, on-demand simulations ([11]). Unlike traditional systems, LLMs like GPT-4 can simulate dynamic patient behavior, adapt to diverse learner inputs, and provide real-time feedback, potentially transforming how students rehearse complex conversations such as diagnostic delivery ([24]). Indeed, the use of virtual patient software has already shown positive educational effects; studies and systematic reviews report that computer-based simulations can enhance students’ clinical reasoning and skill acquisition and are generally well received by learners ([16]). Now, state-of-the-art generative models like ChatGPT enable highly dynamic patient–doctor conversations that were not previously possible. Educators can leverage these AI-driven virtual patients to offer medical trainees essentially unlimited practice of clinical scenarios without the logistical constraints of scheduling actors or specialized labs ([16]). Such AI-powered simulations can be tailored to specific learning objectives, provide immediate feedback, and ensure standardized yet responsive patient interactions for every student ([16]). In sum, combining time-tested simulation pedagogy with modern AI technology presents a compelling approach to improve diagnostic communication training in a scalable way.

Early experiences with generative AI in medical training are emerging, and the results are promising. Other pilot studies have explored using ChatGPT or related LLMs as conversational partners for clinical simulations. For instance, Scherr et al. demonstrated that ChatGPT-3.5 could successfully generate interactive clinical case simulations, allowing students to practice forming diagnostic impressions and management plans over a full patient encounter ([20]). Notably, the AI-driven cases were able to adapt to learners’ inputs and questions, more closely mimicking real-life dialogues than static text vignettes ([20]). Other interventions have focused specifically on communication skills. One recent pilot study programmed ChatGPT as a virtual patient in a breaking-bad-news scenario (using the SPIKES protocol) and had second-year medical students conduct a difficult diagnosis discussion via text. The pre–post findings were encouraging: students’ self-confidence in communicating with patients—particularly in delivering bad news—increased significantly after the ChatGPT interaction, and their trust in AI as a useful training tool also improved ([2]). Qualitative feedback from that study indicated that learners valued the structured practice and felt the AI exercise helped them understand the patient’s perspective ([2]). Similarly, another experiment using an advanced voice-interactive ChatGPT-4 system as a virtual standardized patient found that medical trainees were satisfied with the realism of the encounter and were enthusiastic about integrating such AI-based practice into their learning ([16]). Participants in that study appreciated the uniform and safe environment the AI patient provided, and many reported the experience as a valuable supplement to their clinical education ([16]). At the same time, students acknowledged the current limitations of AI simulations—for example, a chatbot cannot yet fully convey human emotions or nuanced non-verbal cues. In the breaking-bad-news simulation, learners emphasized that the ChatGPT exercise was a helpful adjunct but not a replacement for real patient interaction, largely due to the lack of genuine emotional exchange and the predictable nature of AI responses ([2]). These early trials underscore both the potential and the challenges of generative AI in communication training. Overall, the evidence to date—though still limited in scope—suggests that AI chatbots can provide meaningful practice that improves learners’ confidence and skills, especially when used as part of a blended educational approach ([2]).

In educational research, pre–post intervention designs are commonly used to evaluate the impact of novel teaching methods on learners’ skills and attitudes. By measuring outcomes before and after an intervention in the same cohort, this design allows for an initial assessment of effectiveness without requiring a separate control group. Such an approach has been applied in communication skills training and has demonstrated clear gains. For example, one program that combined didactics and a simulated patient encounter to teach first-year students how to deliver bad news saw the proportion of students reporting confidence in this skill jump from 32% before training to 91% afterward ([17]). In a similar vein, the ChatGPT-based breaking-bad-news pilot described above used a pre–post survey to show significant improvement in students’ self-rated communication confidence following the AI-facilitated practice session ([2]). Although pre–post studies have inherent limitations (such as lack of a parallel control group and potential response-shift bias), they are invaluable for pilot assessments of educational innovations, helping to determine whether an intervention shows promise and warrants further development or more rigorous testing ([2]; [17]).

In this context, the present study aimed to evaluate the effectiveness of a generative AI-driven simulation tool designed to train diagnostic communication skills in medical students. Through a structured pre–post intervention, we assessed whether repeated simulated conversations with a responsive virtual patient could improve learners’ ability to deliver a diagnosis clearly, accurately, and empathetically. Beyond feasibility, we sought to generate robust evidence on learning outcomes and predictive factors associated with greater improvement. The simulated cases focused on T2DM and were constructed in accordance with the American Diabetes Association’s (ADA) 2025 Standards of Care and the European Association for the Study of Diabetes (EASD) 2023 Guidelines on Patient Communication, ensuring clinical and pedagogical relevance ([1]; [10]). By combining performance-based assessment, cluster analysis, and learner feedback, this study contributes to the growing field of AI-enhanced medical education and supports the integration of generative AI as a pedagogical adjunct for clinical communication training ([21]).

## 2. Materials and Methods

### 2.1. Study Design and Overview

The DIALOGUE (DIagnostic AI Learning through Objective Guided User Experience) study was a prospective, single-arm, pre–post intervention designed to evaluate the effectiveness of a GenAI-based training program on diagnostic communication performance in medical students. The study took place between May and June 2025 at the Facultad de Estudios Superiores Iztacala (FESI), Universidad Nacional Autónoma de México (UNAM). Ethics approval was obtained from the institutional committee (ID: CE/FESI/042025/1922), and all participants provided written informed consent. A target sample size of 30–32 students was selected based on feasibility for a pre–post design with blinded evaluation, consistent with prior AI-based educational studies in medical training ([2]; [17]). This number was considered sufficient to detect large within-subject effect sizes in communication outcomes, assuming 80% statistical power and α = 0.05, as supported by previous pilot simulations.

### 2.2. Participants

Eligible participants were undergraduate medical students from the FESI-UNAM and enrolled in the clinical phase of their MD program (semesters 2 to 7). Convenience sampling was used to recruit students from May to June 2025. Inclusion criteria included the following: (a) enrollment in clinical semesters, (b) completion of at least one core clerkship, (c) no prior formal training in diagnostic communication, and (d) no participation in prior pilot studies involving AI tools. Baseline characteristics—including age, gender, academic semester, GPA, prior use of generative AI tools, digital self-efficacy, empathy (Jefferson Scale of Empathy–Student version), and previous clinical exposure to real patients—were collected via structured questionnaires. The MD program at FES-Iztacala spans six years, including preclinical and clinical phases. The clinical phase begins in year 3 and progresses through structured clerkships and simulation-based training modules.

### 2.3. Study Phases

The study was conducted in three sequential phases: (1) a pre-test diagnostic communication simulation, (2) an asynchronous remote AI-based training intervention, and (3) a post-test diagnostic consultation with a human SP.

#### 2.3.1. Pre-Test Phase

In the pre-test phase, each participant completed two simulated diagnostic encounters, where they were required to communicate a diagnosis of T2DM to a virtual patient powered by a generative AI model (ChatGPT, GPT-4o-mini, release 15 May 2025; temperature = 0.7, max_tokens = 1000). Prior to each simulation, participants were provided with a five-minute written clinical case summary. Interactions were conducted individually, in real time, through audio-based dialogue and without scripted prompts. Each performance was independently evaluated by two blinded clinical raters using a validated 8-domain diagnostic communication rubric (see Section 2.4). All clinical scenarios were co-designed by the research team and faculty members with expertise in endocrinology and medical education. Each case involved a newly diagnosed adult with type 2 diabetes mellitus and incorporated psychosocial variables of moderate complexity, such as denial, family concern, or emotional distress. Both scenarios presented a diagnosis of T2DM but differed slightly in psychosocial framing—Scenario 1 involved a middle-aged patient with minimal emotional reaction, while Scenario 2 portrayed a younger adult with significant concern about long-term complications and lifestyle impact. All participants completed both scenarios in randomized order to mitigate sequencing bias and ensure balanced exposure. The diagnostic criteria, clinical framing, and communication strategies embedded in the scenarios were designed in accordance with international guidelines, including the ADA Standards of Care in Diabetes—2025 and the EASD 2023 Guidelines for Effective Patient Communication in Diabetes Care ([1]; [10]).

#### 2.3.2. Educational Intervention: AI-Based Training

Following the pre-test, participants completed an asynchronous two-part educational intervention delivered remotely:

Module 1: Prompt Engineering Workshop—A 20 min instructional session introducing the principles of effective prompt construction for clinical simulations using generative AI tools.

Module 2: Diagnostic Communication Workshop—A 40 min training video focused on core elements of patient-centered diagnostic delivery, including empathy, emotional regulation, clarity of explanation, and strategies for communicating a T2DM diagnosis.

After completing both modules, each student conducted ten independent simulated diagnostic conversations with ChatGPT acting as a virtual patient. Each scenario was designed with increasing psychosocial complexity and targeted specific communication competencies (e.g., adapting medical language to patient understanding, managing emotional responses, or addressing denial). Simulations were performed in separate conversation threads. At the conclusion of each interaction, participants entered the prompt “FEEDBACK” to trigger an automated reflection by the AI, providing personalized formative guidance in natural language. All simulation links were submitted to the study coordinator and archived for subsequent qualitative analysis.

#### 2.3.3. Post-Test Phase

Seven days after completing the AI-based training, each participant engaged in two live diagnostic consultations with human standardized patients (SPs), each representing one of the two original clinical scenarios. Thus, every participant was delivered two diagnoses during the post-test phase. The encounters lasted approximately 5–7 min each and were conducted in person. Performances were assessed in real time by a third clinical evaluator, blinded to the participant’s prior exposure to the AI intervention. Scoring was based on the same 8-domain rubric used in the pre-test phase to ensure consistency across evaluations. Post-test evaluations were carried out by clinical raters who were licensed physicians with 5 to 15 years of clinical experience and formal teaching appointments in the MD program at FES-Iztacala. All raters underwent structured calibration using a standardized scoring manual and were blinded to participants’ pre-test scores and AI training exposure.

### 2.4. Evaluation Instruments

The primary outcome was the improvement in diagnostic communication competency, assessed using a modified version of the Kalamazoo Essential Elements Communication Checklist (Adapted), originally developed by the American Academy on Communication in Healthcare ([12]). The rubric was adapted for T2DM diagnostic disclosure and included eight domains: (1) opening and rapport, (2) patient-centered history, (3) empathic listening, (4) clarity of explanation, (5) emotional containment, (6) lay language use, (7) shared decision-making, and (8) professionalism. Each domain was scored on a 5-point Likert scale (1 = poor, 5 = excellent). Rubric descriptors were refined through pilot testing with five students not included in the main analysis. Clinical evaluators received structured calibration training using a standardized scoring manual. Inter-rater discrepancies ≥2 points were resolved through consensus discussion or adjudication by a third blinded reviewer.

### 2.5. Data Collection and Analysis

Rubric scores and self-reported measures (confidence and empathy) were collected manually using standardized paper-based forms and subsequently transcribed into a digital spreadsheet. Pre- and post-test scores were paired for each participant using anonymized identifiers. All statistical analyses were performed using R version 4.5.1 (R Foundation for Statistical Computing, Vienna, Austria). The Shapiro–Wilk test was employed to assess the normality of score distributions. Depending on the results, either paired *t*-tests or Wilcoxon signed-rank tests were applied to compare pre- and post-intervention scores at both the total and domain-specific levels. Effect sizes were calculated using Cohen’s d for parametric tests or r for non-parametric comparisons. Inter-rater reliability was evaluated using intraclass correlation coefficients (ICCs; two-way random effects model, absolute agreement).

### 2.6. Qualitative Analysis of AI Feedback

AI-generated feedback from the training phase was analyzed through inductive thematic analysis. Two independent researchers manually coded the feedback using Microsoft Excel (Microsoft Corporation, Redmond, WA, USA), following an iterative, line-by-line approach. Codes were then grouped into themes through constant comparison. Discrepancies in coding were resolved through discussion until consensus was reached. The process aimed to identify core patterns in learners’ reflections and AI-generated suggestions.

### 2.7. Use of Generative AI in the Study

Generative AI was utilized in two distinct capacities within the study: (1) ChatGPT (GPT-4o-mini; OpenAI, San Francisco, CA, USA) served as a virtual simulated patient during the training phase; and (2) the same model was prompted to generate natural language feedback following each simulated consultation. No model fine-tuning or external training was conducted. Safety settings, token limits, and API parameters were configured in accordance with OpenAI’s safety guidelines (version 2.4). Generative AI was not used in the writing, editing, or translation of the manuscript text.

### 2.8. Statistical Analysis

All statistical analyses were conducted using R version 4.5.1 (R Foundation for Statistical Computing, Vienna, Austria). A two-tailed *p*-value < 0.05 was considered statistically significant. Baseline characteristics were summarized using descriptive statistics. Normality of continuous variables was assessed via the Shapiro–Wilk test. Paired comparisons between pre- and post-intervention scores (total and domain-specific) were performed using paired *t*-tests or Wilcoxon signed-rank tests, with effect sizes reported as Cohen’s d or rank-based r, respectively. Improvement scores (Δ-scores) were calculated as the difference between post-test and pre-test rubric totals. To identify predictors of improvement, we fitted multiple linear regression models, including baseline empathy, self-efficacy, GPA, gender, prior AI use, and other relevant covariates. Model selection followed stepwise procedures using the Akaike Information Criterion (AIC), and multicollinearity was assessed using variance inflation factors (VIFs). Model assumptions were verified via residual plots. Cluster analysis (k-means, k = 3) was performed on standardized baseline variables to identify latent learner profiles. Clusters were compared on Δ-scores using ANOVA or Kruskal–Wallis tests with Tukey or Dunn’s post hoc corrections. Correlation analyses (Pearson or Spearman) were used to explore associations between baseline traits (e.g., empathy, motivation) and performance outcomes. Rubric reliability was assessed through Cronbach’s alpha (internal consistency) and intraclass correlation coefficients (ICC, two-way random effects, absolute agreement) for inter-rater agreement. Sensitivity analyses excluded participants with high baseline scores or incomplete training engagement. No data imputation was performed.

## 3. Results

### 3.1. Participant Flow and Baseline Characteristics

#### 3.1.1. Participant Flow

Of the 41 medical students assessed for eligibility, 32 met the inclusion criteria and agreed to participate (Figure 1). All 32 students completed the baseline questionnaire and both pre-test diagnostic consultations with an AI-based virtual patient. Subsequently, all enrolled participants engaged in and completed the asynchronous AI remote training module. However, two students (6.3%) were lost to follow-up and did not attend the scheduled human standardized patient (SP) post-test. Therefore, 30 participants (93.8%) completed the full study protocol and were included in both the intention-to-treat (ITT) and per-protocol analyses.

#### 3.1.2. Baseline Characteristics

Baseline characteristics for the enrolled cohort (n = 32) are summarized in Table 1. The mean age was 21.1 ± 4.1 years (range: 19–43), with 22 participants identifying as female (73.3%) and 8 as male (26.7%). Most students were in the fourth clinical semester (63.3%), with smaller proportions in semester 2 (13.3%), semester 6 (6.7%), and semester 7 (16.7%). No participant had previously received formal instruction in diagnostic communication, and none had completed an Objective Structured Clinical Examination (OSCE). Over two-thirds of students (70%) reported having ≥10 prior interactions with ChatGPT or similar large language models (LLMs), and 40% had previous experience communicating clinical findings to real patients. Digital self-efficacy was rated as high or very high by 43.3%, medium by 53.3%, and low by 3.3% of students. Laptops or desktop computers were the most frequently used devices for simulation (73.3%), followed by tablets and smartphones. Internet quality was self-rated as “good” (56.7%), “medium” (33.3%), or “poor” (10%). The median self-reported motivation to participate was 9 on a 10-point scale. Confidence in core communication skills varied: it was highest for explaining laboratory results (mean = 3.3 ± 0.7) and lowest for performing teach-back (mean = 2.3 ± 0.8). The mean total score on the Jefferson Scale of Empathy was 116 ± 15.

### 3.2. Baseline Diagnostic-Communication Performance

At baseline, the mean total rubric score was 48.83 ± 9.65 for Scenario 1 and 51.10 ± 9.82 for Scenario 2, showing a modest but statistically significant difference (*p* = 0.05). This suggests that some participants may have experienced procedural gains or growing familiarity with the simulation format between the two baseline encounters. Despite this overall difference, no significant variation was observed at the domain level (all *p* > 0.05), indicating general consistency across the eight communication domains. However, item-level comparisons revealed significant differences in a few key subskills. Participants scored higher in Scenario 2 when assessed on their ability to assess the daily-life impact of diabetes (Item 4.3, *p* = 0.02), to check patient understanding through teach-back (Item 5.3, *p* = 0.01), and to close the consultation with empathy and follow-up planning (Item 7.3, *p* = 0.02). Additionally, the ability to provide a final summary (Item 7.1) approached significance (*p* = 0.05), suggesting a mild improvement in closure behavior during the second encounter. These differences likely reflect initial learning effects or adjustment to the AI-simulated environment rather than true intervention effects, given that no formal training had yet occurred. Across both scenarios, most participants demonstrated limited baseline communication proficiency. Based on total scores, 63% (19/30) of students were classified as low performers, 33% (10/30) as intermediate, and none as high performers. The lowest baseline scores were consistently observed in the subdomains of teach-back and goal negotiation. In contrast, active listening (Item 3.2) received the highest mean score (4.00 ± 0.18 in Scenario 1). However, this rating should be interpreted with caution, as it may reflect the structural nature of AI-mediated conversations. Unlike human interactions—where overlaps and interruptions are common—dialogues with ChatGPT occur in a turn-based format, which may have inadvertently facilitated uninterrupted responses and inflated perceived listening quality. A radar plot comparing mean domain scores between the two pre-test scenarios (Figure 2) visually confirms the similarity in participants’ communication profiles, with only minor fluctuations between domains. A detailed breakdown of scores by item and scenario is provided in Table 2.

### 3.3. Post-Intervention Diagnostic Performance and Pre–Post Comparison

#### 3.3.1. Post-Test Performance Across Scenarios

After completing the AI-based intervention, participants underwent two live diagnostic consultations with standardized patients. The mean total rubric scores were 84.5 ± 17.8 for Scenario 1 and 88.9 ± 17.4 for Scenario 2 (*p* = 0.08), with no statistically significant difference between scenarios. As shown in Table 3 and Figure 3, domain-level analyses revealed consistent performance across both encounters. Slight variations emerged in patient greeting (D2.1, *p* = 0.01), active listening (D3.2, *p* = 0.01), and diabetes specific plan explanation (D8.2, *p* = 0.02), but overall post-test profiles were comparable. The spider plot in Figure 4 illustrates the near-overlapping domain scores between scenarios, suggesting internal consistency and performance stability across post-test cases.

#### 3.3.2. Pre–Post Intervention Gains

When comparing pre- and post-intervention results, participants demonstrated substantial and statistically significant improvements across all rubric domains. The total score increased from 49.96 ± 9.72 to 86.70 ± 17.56 (Δ = 36.74, 95% CI: 31.39 to 42.09, *p* < 0.001; Cohen’s d = 2.58), representing a very large effect size. As shown in Table 4, every domain showed meaningful gains, with the largest improvements observed in “Opening” (Δ = 1.92, d = 2.68), “Closure” (Δ = 1.79, d = 2.07), and “Diabetes specific explanation” (Δ = 2.06, d = 2.95). The most improved subitems included teach-back (Item 5.3, Δ = 2.12), goal negotiation (6.3, Δ = 1.58), and empathetic closure (7.3, Δ = 1.77), all with large effect sizes. Figure 4 presents a radar plot overlaying pre- and post-intervention domain means, visually confirming significant gains across all communication areas. In parallel, Figure 5 displays a violin box plot contrasting the total rubric score distribution before and after training, highlighting both the upward shift in central tendency and reduced score dispersion after the intervention. Together, these findings indicate that the generative AI-based training module produced broad and meaningful enhancements in students’ diagnostic communication competencies, spanning both structural and affective elements of clinical dialogue.

### 3.4. Predictors of Improvement

To identify baseline factors associated with the magnitude of improvement in diagnostic-communication scores, we fitted a multiple linear regression model with the Δ-score (post–pre total rubric score) as the dependent variable. Candidate predictors were age, gender, school term, GPA, previous interaction with real patients (yes/no), baseline empathy (Jefferson Scale total score), digital self-efficacy (1–5 scale), self-reported motivation (1–10 scale), and prior use of ChatGPT or other LLMs (≥10 interactions vs. <10). The overall model was statistically significant (R^2^ = 0.43, adjusted R^2^ = 0.37; F(6, 23) = 4.13, *p =* 0.004), explaining approximately 37% of the variance in Δ-scores. Two variables emerged as independent predictors: baseline empathy was negatively associated with improvement (β = −0.41, SE = 0.13; 95% CI: −0.68 to −0.14; *p* = 0.005; Figure 6E), indicating that students with lower initial empathy scores tended to achieve larger gains, likely due to a ceiling effect among highly empathic learners. Conversely, digital self-efficacy was positively associated with improvement (β = 0.35, SE = 0.14; 95% CI: 0.07–0.63; *p* = 0.016; Figure 6D), suggesting that greater confidence with digital tools enhanced the effectiveness of GenAI training. Gender showed a marginal effect (*p* = 0.044; Figure 6G) in the bivariate analysis, with female students demonstrating higher median Δ-scores; however, this difference was no longer significant after adjusting for other covariates. No significant associations were observed for age (Figure 6A), school term (Figure 6B), GPA (Figure 6C), motivation (Figure 6F), prior patient contact (Figure 6H), or previous LLM use. Overall, these results indicate that learners entering the training with lower empathy yet stronger digital self-efficacy derived the greatest incremental benefit from GenAI-mediated communication practice, highlighting the combined influence of emotional baseline and technological readiness on learning outcomes. Figure 6 visualizes these relationships through scatterplots (panels A–F) and boxplots (panels G–H), underscoring the heterogeneity of individual learning trajectories.

### 3.5. Learner Profiles and Cluster Patterns

To examine differential response patterns, we performed unsupervised k-means clustering (k = 3) based on participants’ domain-level rubric scores before and after the intervention. The optimal k was confirmed with the elbow method and inspection of within-cluster sum-of-squares. Three discrete learner profiles emerged. Cluster A (n = 10) achieved the greatest mean improvement (Δ = 58.7 ± 10.2), followed by Cluster B (n = 12; Δ = 33.4 ± 11.5) and Cluster C (n = 8; Δ = 16.2 ± 9.3). Between-cluster differences in total Δ-score and every domain-specific Δ-score were significant (*p* < 0.001; Table 5), indicating meaningful heterogeneity in learning gains. Figure 7 shows domain-specific Cohen’s d values, all in the moderate-to-large range. The largest effects occurred in Domain 8 (diabetes specific explanation, d = 2.95), Domain 2 (opening the encounter, d = 2.68), and Domain 5 (information sharing, d = 2.17). The heat map and dendrogram in Figure 8 depict these patterns: Cluster A displays uniformly high gains across all eight domains, whereas Cluster C shows marginal change, especially in “patient perspective” and “shared decision-making”. Baseline trait inspection revealed that Cluster A learners started with lower empathy but higher digital self-efficacy, consistent with predictors identified in Section 3.4; Cluster C showed the opposite pattern (higher empathy, lower self-efficacy). A scatterplot of baseline total rubric score versus Δ-score (Figure 9) demonstrated a weak inverse relationship (r = −0.22, *p* = 0.24), supporting a compensatory effect in which students with lower starting performance realised larger gains. Collectively, these findings suggest that GenAI-mediated practice may help narrow performance gaps by especially benefiting learners who begin with weaker communication skills yet possess the technological confidence to exploit AI feedback.

### 3.6. Post-Test Surveys

Upon completion of the post-test phase, all 30 students responded to a structured post-intervention questionnaire assessing perceived knowledge acquisition, self-efficacy in diagnostic communication, and the perceived usefulness of the GenAI training. The overall self-reported knowledge score averaged 4.2 ± 0.6 on a five-point Likert scale, while self-efficacy in delivering a diagnosis improved from a baseline mean of 2.9 ± 0.8 to 4.3 ± 0.7. Notably, 93.3% (28/30) of participants rated the GenAI practice sessions as “very useful” or “extremely useful” for reinforcing communication skills, and 86.7% (26/30) expressed willingness to integrate AI-based simulation into future clinical training. Open-ended responses highlighted three recurring themes: (1) increased comfort navigating emotionally charged conversations, (2) appreciation for structured feedback in natural language, and (3) a desire for broader case variety and real-time faculty debriefing in future iterations. Additionally, both evaluator and SP post-revelation surveys (administered after disclosure of AI use in training) revealed no detection bias; only one out of six evaluators and eight out of 24 SPs suspected that students had undergone prior AI-based preparation. Following disclosure, both groups acknowledged that the students’ communication had exceeded their expectations.

### 3.7. Adverse Events and Missing Data

No adverse events or psychological distress were reported by participants during or after the intervention. Two students who completed the full AI-based remote training withdrew before the post-test simulation due to scheduling conflicts and were excluded from final analyses. No technical failures occurred during pre- or post-test evaluations. Two minor discrepancies in rubric scoring (≥2-point difference between evaluators) were resolved through consensus with a third blinded reviewer. All data points from the 30 remaining participants were complete and included in the final analyses.

## 4. Discussion

This study provides robust evidence that a deliberately scaffolded educational intervention powered by generative artificial intelligence (GenAI) can significantly improve diagnostic communication skills among undergraduate medical students. The intervention yielded a substantial within-subject gain (Δ = +36.7 points on a 115-point rubric), with large effect sizes across all eight assessed domains. These results reflect a broad-based enhancement of both cognitive-structural and emotional-affective communication competencies rather than isolated improvements in discrete tasks. T2DM was intentionally selected as the diagnostic focus of all scenarios due to its high clinical prevalence, familiarity among medical students, and capacity to elicit both cognitive processing and empathic engagement. This choice likely facilitated student immersion and increased the ecological validity of the simulations, allowing participants to focus on communication delivery rather than clinical unfamiliarity. By anchoring the intervention in a universally relevant and emotionally resonant condition, the study maximized its potential to reveal true gains in communicative competence.

Interestingly, 73% of the participants were female, which closely reflects the actual gender distribution in the MD program at FES-Iztacala, where approximately 70% of enrolled students are women. Thus, the gender composition of the sample aligns with institutional demographics rather than selection bias. While female students are often reported to exhibit higher empathy scores and greater receptiveness to communication training, gender showed only a marginal effect on improvement in our multivariate model. Nonetheless, future studies may explore whether gender-related attitudes toward AI tools or communication style mediate training responsiveness. Importantly, our findings advance the current literature by moving beyond student self-report measures toward objective, evaluator-blinded behavioral assessments. Prior studies exploring LLM applications in medical education—such as [23] ([23]) and [2] ([2])—have primarily reported increased learner confidence following GenAI-assisted simulations but lacked external validation through SP encounters or blinded human scoring ([2]; [23]). In contrast, our study combines blinded, domain-based performance assessments with Spanish-language AI-patient dialogues, offering a culturally contextualized and psychometrically rigorous evaluation framework. Furthermore, while Webb et al. employed automated scoring metrics and Chiu et al. focused on English-language communication exercises, our design uniquely integrates live standardized patients, a validated rubric, and a focus on real-time diagnostic disclosure in a Latin American medical context ([2]; [23]). These methodological distinctions underscore the added value of our findings in terms of generalizability and educational depth. To our knowledge, this is the first study to generate such triangulated, rubric-based evidence in Spanish-speaking medical students, within a Latin American setting ([12], [13]).

Several pedagogical mechanisms may underlie the observed gains. First, the GenAI platform provided asynchronous, low-stakes opportunities for deliberate practice—a well-established driver of communication skill acquisition in simulation-based education. Second, immediate feedback in natural language allowed for dynamic self-correction without requiring constant faculty oversight.

Thematic analysis of over 300 AI-generated responses revealed alignment with communication best practices (e.g., “organize your explanation,” “validate patient concerns,” “check for understanding”), echoing established heuristics described by ([15]). These effects mirror findings from large-scale meta-analyses on simulation-based education in healthcare, such as that carried out by [4] ([4]), which underscore the importance of repeated practice and structured feedback in skill acquisition ([4]). Third, the virtual patient’s emotionally neutral tone likely created a psychologically safe space, encouraging students to experiment with empathetic strategies without fear of judgment—an effect that mirrors findings from early simulated patient training ([9]).

Regression modeling confirmed an inverse relationship between baseline empathy (measured via the Jefferson Scale) and gains—students with lower initial empathy achieved larger improvements. Simultaneously, students with higher digital self-efficacy benefited more from the GenAI training. These predictors jointly explained over one-third of the variance in learning gains and resonate with constructs from the technology acceptance model ([5]), where perceived usability and personal relevance enhance engagement with digital tools. Interestingly, prior ChatGPT use, motivation level, and GPA were not significant predictors, suggesting that emotional and technological readiness may outweigh cognitive or experiential variables when learning with AI. These results carry practical implications: baseline learner profiles may serve as a foundation for tailoring feedback scaffolding, simulation complexity, or engagement thresholds in future GenAI deployments.

Our unsupervised clustering further highlighted the heterogeneity of learning trajectories. Three learner profiles were identified, with “Cluster A” students—characterized by high empathy and self-efficacy—demonstrating nearly fourfold higher gains compared to their peers. These findings suggest that GenAI is not universally transformative but particularly beneficial for specific learner subtypes. Future adaptive systems could dynamically adjust training sequences based on real-time profiling, maximizing pedagogical efficiency while personalizing learning trajectories.

From a broader curricular standpoint, our findings suggest that GenAI tools—if properly contextualized—can be feasibly integrated into clinical communication training in Mexican medical schools. Faculty shortages, limited access to standardized patients, and rigid curricula pose barriers to consistent development of communication skills. GenAI platforms, especially those adapted to local languages and clinical realities, offer a scalable, low-cost complement. However, successful adoption will require both technical infrastructure and institutional culture change. Some educators may view AI as a threat to pedagogical authority, while others may lack the AI literacy needed to evaluate feedback quality. Addressing this will require deliberate faculty training, transparency in AI decision logic, and curricular frameworks that combine human and machine feedback in meaningful ways.

Looking forward, the evolution of LLMs—such as the expected GPT-4.5 or GPT-5—may radically expand the capabilities of virtual patients. Upcoming models may integrate real-time voice, emotion recognition, and adaptive personas, allowing for multimodal simulation of complex encounters. These advances may offer unprecedented fidelity in diagnostic communication training. However, ethical challenges remain. Overreliance on AI feedback may desensitize learners or reinforce mechanical communication patterns. There is also a risk of depersonalization if students generalize from algorithmic interactions to real patient care. To mitigate these risks, GenAI tools must be embedded within reflective, supervised curricula that cultivate critical thinking, emotional sensitivity, and context-aware communication.

Despite its strengths, this study has several limitations. First, the absence of a control group limits causal inference. However, this was an intentional design choice for this first-phase feasibility and efficacy assessment of the DIALOGUE program. The magnitude and distribution of pre–post gains, combined with evaluator blinding and objective performance metrics, help mitigate—but do not eliminate—concerns about internal validity. Future iterations of the program will include a randomized controlled trial to more rigorously compare GenAI-based training against traditional simulation and didactic instruction. Subsequent phases of the DIALOGUE program will incorporate larger, multicenter samples to enhance statistical power and generalizability. Although evaluator blinding was implemented, participants were aware of the study phase, which may have introduced performance expectancy bias. Second, the short duration precludes conclusions about long-term retention or clinical transfer. Longitudinal follow-up will be incorporated in future studies to assess the persistence of skill gains over time. Future studies may also consider integrating standardized instruments such as the SEGUE framework to enhance transparency and facilitate comparison across educational settings.

Our intervention was deployed within a single institution, potentially limiting generalizability. Although the GenAI platform functioned reliably in Spanish, subtle limitations in semantic nuance or cultural appropriateness may have affected feedback quality. The natural language feedback from the generative AI was not independently validated by clinical educators, which limits conclusions about its educational adequacy. Additionally, while we standardized system prompts and temperature settings, the inherent stochasticity of LLMs introduces minor variations that may affect reproducibility. Future research should explore the impact of AI randomness, develop seed-logging protocols, and benchmark AI feedback against expert commentary.

Finally, while this study provides promising initial evidence of effectiveness, it does not compare GenAI-based feedback to traditional faculty-led instruction nor does it evaluate downstream clinical outcomes such as patient satisfaction or diagnostic accuracy. These remain important avenues for future research. Thus, caution is warranted when interpreting these findings as definitive. Nevertheless, our findings support the role of GenAI as a potentially scalable and psychometrically sound adjunct—not a replacement—for clinical communication training. Although the present intervention centered on T2DM, the framework may be adaptable to other specialties such as pediatrics, psychiatry, or gynecology. Future iterations should investigate domain-specific adaptations and their differential impact on communication skill acquisition.

## 5. Conclusions

This study demonstrates that generative AI can serve as a reliable, scalable tool to enhance diagnostic communication skills in medical students. A short, asynchronous GenAI-based intervention led to significant, domain-wide improvements, particularly among students with higher baseline empathy and digital self-efficacy. While not a replacement for human teaching, GenAI offers a promising pedagogical adjunct, especially in resource-limited settings. Future work should focus on long-term outcomes, integration into curricula, and alignment with ethical, cultural, and clinical standards.

## Figures and Tables

**Figure 1 ejihpe-15-00152-f001:**
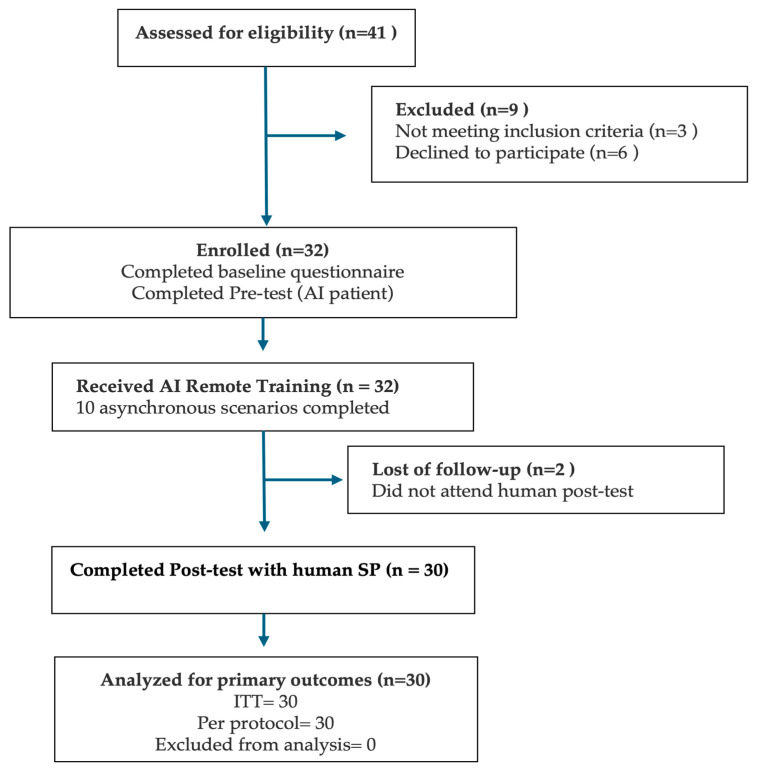
Participant flow through the DIALOGUE pre–post study, based on the 2025 CONSORT diagram. The figure outlines the number of students assessed for eligibility, enrolled, exposed to the AI-based remote training intervention, and completing the SP post-test. Final analyses were conducted on all participants who completed the protocol ([8]).

**Figure 2 ejihpe-15-00152-f002:**
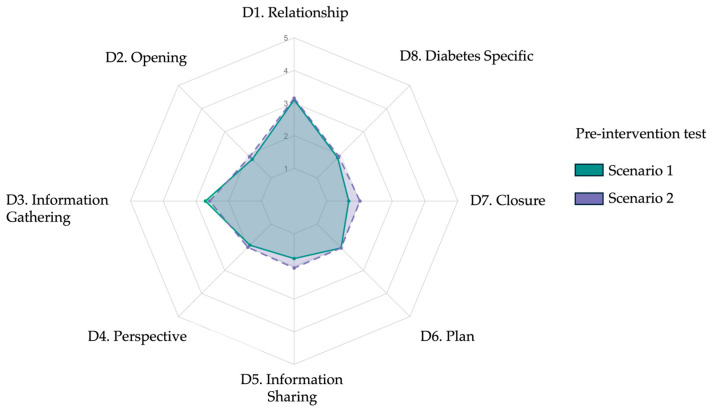
Radar plot comparing mean baseline diagnostic communication scores across eight rubric domains for Scenario 1 and Scenario 2. Each axis represents one domain from the adapted Kalamazoo rubric: (D1) relationship and rapport, (D2) opening the encounter, (D3) information gathering, (D4) exploring patient perspective, (D5) information sharing, (D6) plan negotiation, (D7) closure, and (D8) diabetes specific explanation. Colored lines indicate mean Likert scale scores (range 1–5): green for Scenario 1 and purple for Scenario 2. Pre-intervention profiles were broadly similar across domains.

**Figure 3 ejihpe-15-00152-f003:**
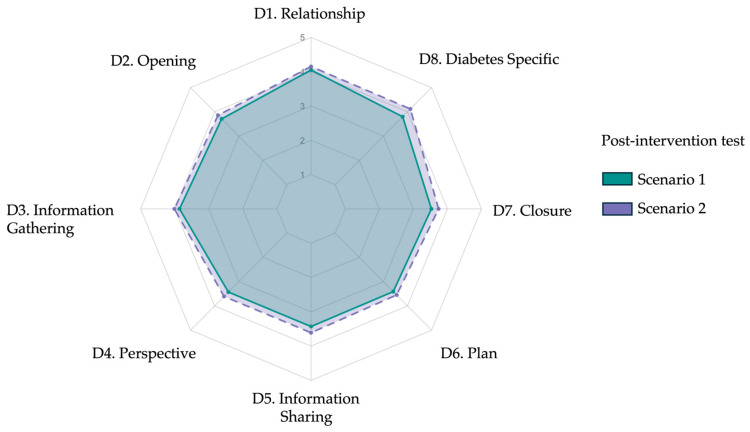
Radar plot comparing mean post-intervention diagnostic communication scores across eight rubric domains for Scenario 1 and Scenario 2. Each axis represents one domain from the adapted Kalamazoo rubric: (D1) relationship and rapport, (D2) opening the encounter, (D3) information gathering, (D4) exploring patient perspective, (D5) information sharing, (D6) plan negotiation, (D7) closure, and (D8) diabetes specific explanation. Colored lines indicate mean Likert scale scores (range 1–5): green for Scenario 1 and purple for Scenario 2. Post-intervention profiles show consistent improvement across domains.

**Figure 4 ejihpe-15-00152-f004:**
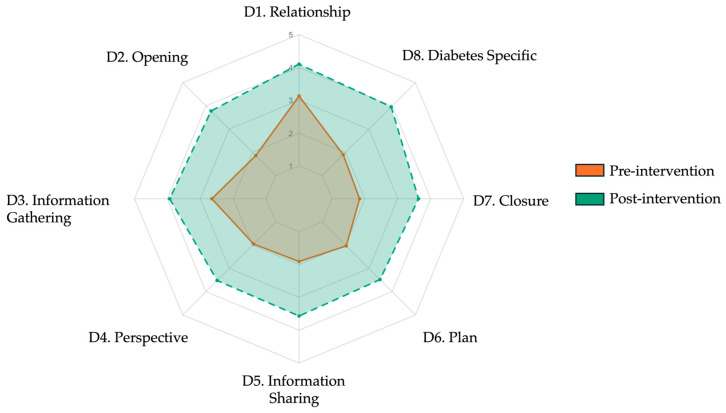
Radar plot comparing mean post-intervention diagnostic communication scores across eight rubric domains for Scenario 1 and Scenario 2. Each axis represents one domain from the adapted Kalamazoo rubric: (D1) relationship and rapport, (D2) opening the encounter, (D3) information gathering, (D4) exploring patient perspective, (D5) information sharing, (D6) plan negotiation, (D7) closure, and (D8) diabetes specific explanation. Colored lines indicate mean Likert scale scores (range 1–5): orange for Scenario 1 and green for Scenario 2. Post-intervention profiles were broadly consistent across scenarios.

**Figure 5 ejihpe-15-00152-f005:**
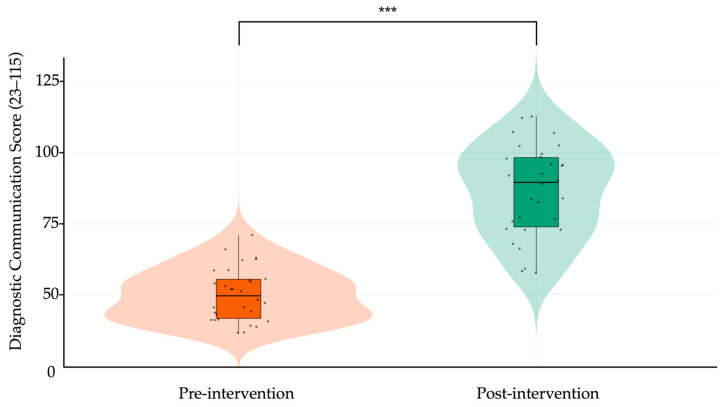
Violin plot comparing pre- and post-intervention diagnostic communication scores. Each dot represents an individual participant’s total score (range: 23–115), derived from the adapted Kalamazoo rubric. Violin plots illustrate the distribution and density of scores; overlaid boxplots show the median (horizontal bar) and interquartile range (box). Vertical lines indicate score range (minimum–maximum). Orange denotes pre-intervention scores; green denotes post-intervention scores. Asterisks (***) indicate statistically significant improvement after the intervention (paired-samples *t*-test, *p* < 0.001).

**Figure 6 ejihpe-15-00152-f006:**
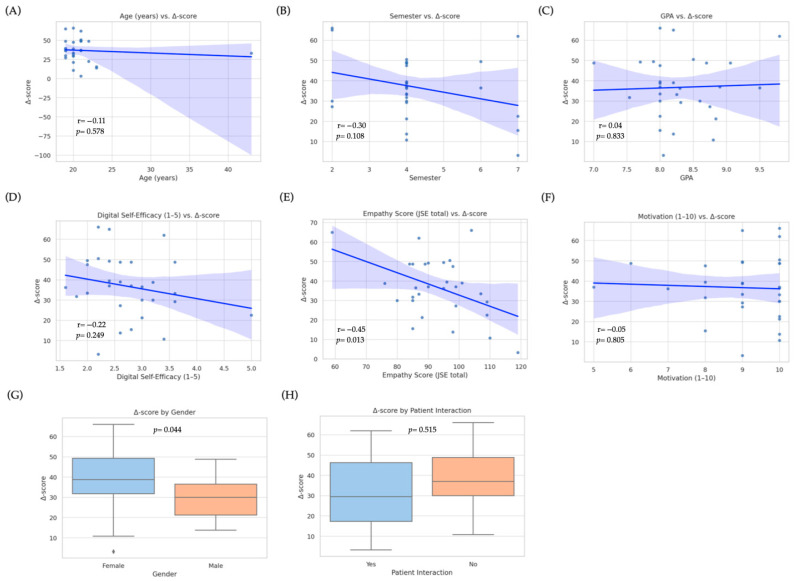
Associations between baseline predictors and improvement in diagnostic-communication scores (Δ-score) among medical students (n = 30). Scatterplots (top and middle rows) display bivariate Pearson correlations between continuous baseline variables and the change in total rubric score (Δ-score = post−pre): (**A**) age (years), (**B**) school term, (**C**) grade point average (GPA), (**D**) digital self-efficacy (1–5 scale), (**E**) empathy score (Jefferson Scale of Empathy, JSE), and (**F**) self-reported motivation (1–10 scale). Solid blue lines represent linear regression trends; shaded areas show 95% confidence intervals. Box-and-whisker plots (bottom row) compare Δ-scores by (**G**) gender (female vs. male) and (**H**) prior patient interaction (yes vs. no). Each dot denotes an individual student. Only higher baseline empathy was significantly associated with lower Δ-scores in the bivariate analysis (R^2^ = −0.45, *p* = 0.013); gender showed a marginal difference (*p* = 0.044). No significant bivariate correlations were observed for age, school term, GPA, digital self-efficacy, motivation, or prior patient contact. (In the multivariate model reported in Section 3.4, digital self-efficacy emerged as an additional positive predictor of improvement.).

**Figure 7 ejihpe-15-00152-f007:**
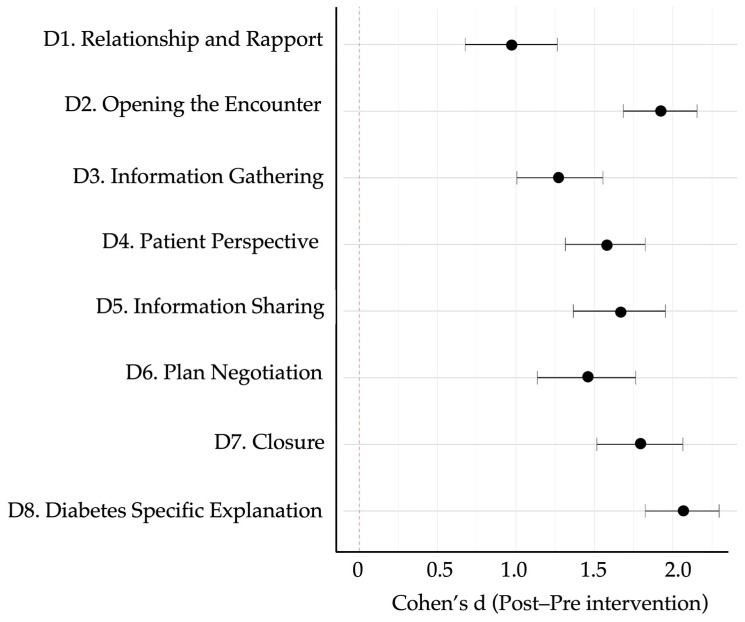
Forest plot showing effect sizes (Cohen’s d) across the eight diagnostic communication domains. All values reflect the magnitude of improvement from pre- to post-intervention. Dots represent the mean effect size per domain; horizontal lines indicate 95% confidence intervals. All domains demonstrated moderate to large effects, with the highest observed in D8 (diabetes specific explanation), D2 (opening), and D5 (information sharing).

**Figure 8 ejihpe-15-00152-f008:**
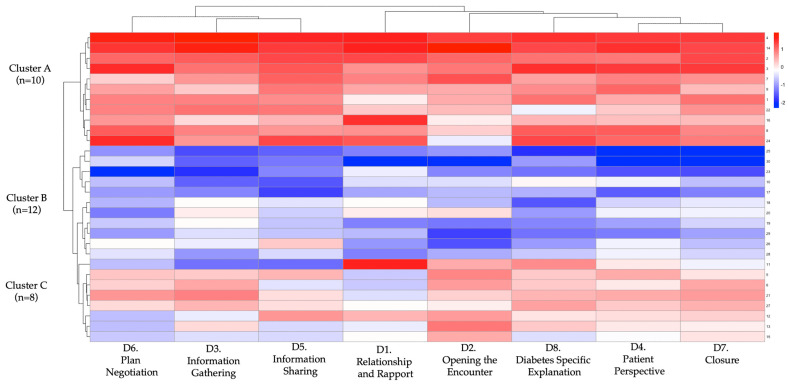
Cluster heatmap and hierarchical dendrogram of learner profiles based on post-intervention diagnostic communication performance. Each row represents an individual student (N = 30), and each column represents one of the eight communication domains from the adapted Kalamazoo rubric: (D1) relationship and rapport, (D2) opening the encounter, (D3) information gathering, (D4) patient perspective, (D5) information sharing, (D6) plan negotiation, (D7) closure, and (D8) diabetes specific explanation. Scores were standardized (z-score transformation) to enable clustering and visualization. A hierarchical cluster analysis using Euclidean distance and complete linkage identified three distinct learner profiles: Cluster A (n = 10; high responders), Cluster B (n = 12; intermediate responders), and Cluster C (n = 8; low responders). The color gradient reflects standardized performance: red indicates higher relative scores and blue indicates lower relative scores across each domain. This visualization allows comparison of performance patterns across individuals and highlights consistent strengths or deficits within clusters.

**Figure 9 ejihpe-15-00152-f009:**
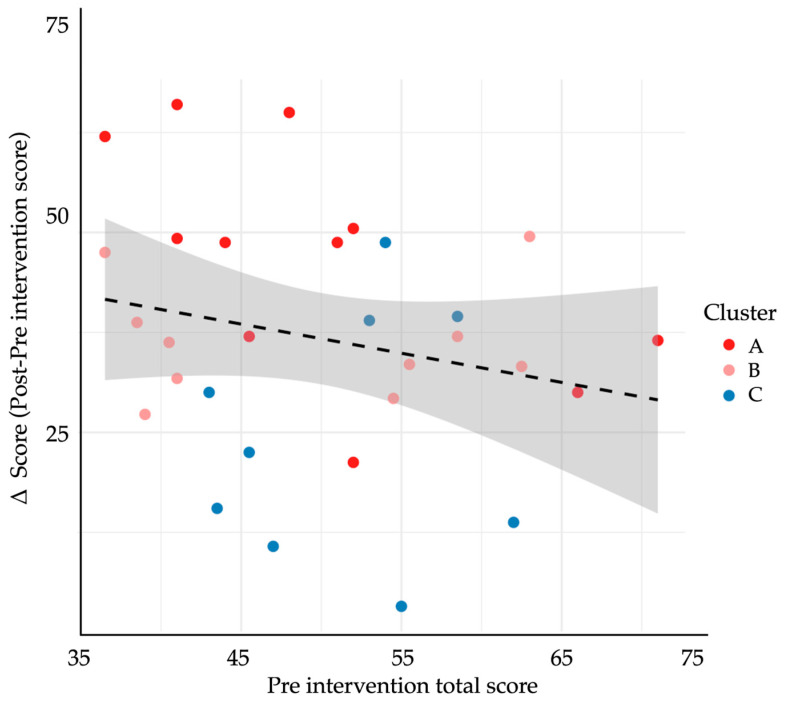
Association between baseline communication performance and improvement following the GenAI-based intervention. The scatterplot displays the relationship between pre-intervention rubric scores (*x*-axis) and Δ-scores (*y*-axis), calculated as post−pre total rubric score. Each point represents one student (N = 30), color-coded by cluster membership (Cluster A = red, B = pink, C = blue). The dashed regression line represents the best linear fit, and the shaded area indicates the 95% confidence interval. A negative trend was observed, suggesting that students with lower initial scores experienced greater improvements.

**Table 1 ejihpe-15-00152-t001:** Baseline sociodemographic, academic, and digital profile of enrolled medical students (n = 30).

Variable	Category/Units	N (%) or Mean ± SD
Age	Years	21.1 ± 4.1
Sex	Female	22 (73.33%)
	Male	8 (26.66%)
Clinical semester	2	4 (13.33%)
	4	19 (63.33%)
	6	2 (6.66%)
	7	5 (16.66%)
Cumulative GPA	0–10	8.31 ± 0.50
Prior formal course in diagnostic communication	Yes	0 (0%)
Prior ECOE completed	Median (IQR)	0 (0%)
Prior experience with real patients	Yes	12 (40.00%)
Prior ChatGPT/LLM use	≥10 interactions	21 (70%)
	<10 interactions	9 (30%)
	Never	1 (3.33%)
Digital self-efficacy †	Very high/high	13 (43.33%)
	Medium	16 (53.33%)
	Low	1 (3.33%)
Device most used for simulation	Laptop/PC	22 (73.33%)
	Tablet	4 (13.33%)
	Smartphone	4 (13.33%)
Self-rated internet quality	Good	17 (56.66%)
	Medium	10 (32.25%)
	Poor	3 (10%)
Motivation score (1–10)	0–10	9
Self-confidence score, 1–5	Explain lab results	3.3 ± 0.7
	Explain DM criteria	3.0 ± 0.8
	Convey bad news empathetically	3.2 ± 0.8
	Teach-back	2.3 ± 0.8
	Anxiety reduction	2.4 ± 0.7
Jefferson Scale of Empathy, total (20–140)	-	116 ± 15

† Self-efficacy categories derived from a 5-point Likert scale (very low = 1, very high = 5).

**Table 2 ejihpe-15-00152-t002:** Baseline diagnostic-communication scores obtained during the two pre-test scenarios (n = 30).

Item	Domain/Skill (Abridged)	Scenario 1 Mean ± SD	Scenario 2 Mean ± SD	*p*
1	Relationship	3.10 ± 0.58	3.15 ± 0.55	0.50
1.1	Eye contact and open posture	3.86 ± 0.73	3.83 ± 0.79	0.74
1.2	Friendly body language	3.46 ± 0.86	3.56 ± 0.72	0.41
1.3	Self-introduction and role	2.00 ± 1.16	2.06 ± 1.20	0.70
2	Opening	1.81 ± 0.71	1.92 ± 0.69	0.22
2.1	Greeting and ID verification	1.73 ± 0.63	1.63 ± 0.71	0.18
2.2	States purpose of visit	1.90 ± 0.84	2.13 ± 0.89	0.06
2.3	Explores patient expectations	1.80 ± 0.98	2.00 ± 0.94	0.22
3	Information gathering	2.71 ± 0.64	2.58 ± 0.69	0.12
3.1	Uses open questions	2.00 ± 1.01	2.30 ± 0.99	0.76
3.2	Listens without interrupting	4.00 ± 0.18	3.90 ± 0.54	0.32
3.3	Summarizes to confirm	2.13 ± 1.04	1.83 ± 1.11	0.08
4	Patient perspective	1.91 ± 0.85	2.00 ± 0.84	0.36
4.1	Elicits beliefs about illness	1.80 ± 1.06	1.70 ± 1.02	0.57
4.2	Explores worries/concerns	2.13 ± 0.97	1.96 ± 1.03	0.23
4.3	Assesses daily-life impact	1.8 ± 0.92	2.33 ± 1.12	0.02
5	Information sharing	1.76 ± 0.76	2.05 ± 0.83	0.18
5.1	Explains diagnosis in plain language	1.80 ± 1.06	1.70 ± 1.02	0.57
5.2	Uses visual aids/examples	2.13 ± 0.97	1.96 ± 1.03	0.23
5.3	Checks understanding (teach-back)	1.80 ± 0.92	2.33 ± 1.12	0.01
6	Plan negotiation	2.03 ± 0.65	2.03 ± 0.53	1.00
6.1	Discusses therapeutic options	2.36 ± 0.85	2.10 ± 0.99	0.07
6.2	Involves patient in decisions	2.06 ± 0.90	2.16 ± 0.74	0.44
6.3	Negotiates realistic goals	1.66 ± 0.80	1.83 ± 0.83	0.20
7	Closure	1.67 ± 0.62	2.01 ± 0.59	0.06
7.1	Provides final summary	1.63 ± 0.71	1.86 ± 0.81	0.05
7.2	Checks for residual questions	1.53 ± 0.93	1.86 ± 0.93	0.07
7.3	Closes with empathy and follow-up	1.86 ± 0.97	2.30 ± 0.83	0.02
8	Diabetes specific	1.88 ± 0.56	1.93 ± 0.56	0.73
8.1	Explains lab results (HbA1c)	1.80 ± 0.71	1.90 ± 0.71	0.50
8.2	Provides initial plan and alleviates anxiety	1.96 ± 0.71	1.90 ± 0.75	0.57
Total	Overall score (23–115)	48.83 ± 9.65	51.1 ± 9.82	0.05

Paired-samples *t*-test; Shapiro–Wilk confirmed normality for all difference distributions. Significant differences (*p* < 0.05) observed in Items 4.3, 5.3, and 7.3 (paired *t*-tests).

**Table 3 ejihpe-15-00152-t003:** Baseline diagnostic-communication scores obtained during the two post-test scenarios (N = 30). No statistically significant differences were observed between Scenario 1 and Scenario 2 for any rubric item (paired *t*-tests, all *p* > 0.05).

Item	Domain/Skill (Abridged)	Scenario 1 Mean ± SD	Scenario 2 Mean ± SD	*p*
1	Relationship	4.05 ± 0.63	4.15 ± 0.67	0.36
1.1	Eye contact and open posture	4.15 ± 0.54	4.26 ± 0.86	0.21
1.2	Friendly body language	4.26 ± 0.55	4.26 ± 0.62	1.00
1.3	Self-introduction and role	3.85 ± 0.84	3.81 ± 0.86	0.82
2	Opening	3.71 ± 0.64	3.86 ± 0.81	0.25
2.1	Greeting and ID verification	3.81 ± 0.71	4.35 ± 0.84	0.01
2.2	States purpose of visit	3.68 ± 0.78	3.93 ± 0.96	0.11
2.3	Explores patient expectations	3.63 ± 0.81	3.30 ± 1.04	0.16
3	Information gathering	3.86 ± 0.60	4.01 ± 0.68	0.19
3.1	Uses open questions	3.88 ± 0.67	3.96 ± 0.00	0.56
3.2	Listens without interrupting	4.21 ± 0.31	4.43 ± 0.40	0.01
3.3	Summarizes to confirm	3.50 ± 1.09	3.63 ± 1.12	0.08
4	Patient perspective	3.43 ± 0.62	3.61 ± 0.54	0.31
4.1	Elicits beliefs about illness	3.21 ± 0.78	3.15 ± 0.82	0.79
4.2	Explores worries/concerns	3.91 ± 0.57	3.60 ± 0.53	0.11
4.3	Assesses daily-life impact	3.50 ± 0.75	3.76 ± 0.52	0.15
5	Information sharing	3.43 ± 1.05	3.61 ± 1.04	0.31
5.1	Explains diagnosis in plain language	3.21 ± 1.09	3.15 ± 0.97	0.79
5.2	Uses visual aids/examples	3.91 ± 1.16	3.60 ± 1.10	0.11
5.3	Checks understanding (teach-back)	3.50 ± 1.23	3.76 ± 1.16	0.15
6	Plan negotiation	3.41 ± 0.95	3.55 ± 1.00	0.34
6.1	Discusses therapeutic options	3.53 ± 0.97	3.70 ± 0.90	0.42
6.2	Involves patient in decisions	3.43 ± 1.11	3.55 ± 1.16	0.37
6.3	Negotiates realistic goals	3.26 ± 1.15	3.40 ± 1.10	0.47
7	Closure	3.53 ± 1.11	3.74 ± 0.99	0.14
7.1	Provides final summary	3.30 ± 1.24	3.46 ± 0.93	0.36
7.2	Checks for residual questions	3.62 ± 0.97	3.75 ± 1.02	0.41
7.3	Closes with empathy and follow-up	3.68 ± 1.28	4.01 ± 1.10	0.06
8	Diabetes specific	3.80 ± 0.81	4.12 ± 0.79	0.03
8.1	Explains lab results (HbA1c)	3.81 ± 0.80	4.05 ± 0.71	0.12
8.2	Provides initial plan and alleviates anxiety	3.78 ± 0.97	4.20 ± 0.74	0.02
Total	Overall score (23–115)	84.46 ± 17.75	88.93 ± 17.37	0.08

Paired-samples *t*-test; Shapiro–Wilk confirmed normality for all difference distributions.

**Table 4 ejihpe-15-00152-t004:** Pre–post comparison of diagnostic communication scores across rubric domains (N = 30). Mean scores, mean differences (Δ), 95% confidence intervals, effect sizes (Cohen’s d), and *p*-values are shown for each rubric item comparing pre- and post-intervention performance. All differences were statistically significant (*p* < 0.001, paired *t*-tests).

Item	Domain/Skill (Abridged)	Pre-Intervention Mean ± SD	Post-Intervention Mean ± SD	Δ Mean	95% CI (Δ)	Cohen’s d	*p*
1	Relationship	3.13 ± 0.56	4.10 ± 0.65	0.97	0.74–1.20	1.59	0.001
1.1	Eye contact and open posture	3.85 ± 0.75	4.20 ± 0.60	0.35	0.09–0.61	0.51	0.004
1.2	Friendly body language	3.51 ± 0.79	4.26 ± 0.58	0.75	0.49–1.01	1.08	0.001
1.3	Self-introduction and role	2.03 ± 1.19	3.83 ± 0.84	1.80	1.41–2.19	1.74	0.001
2	Opening	1.86 ± 0.70	3.78 ± 0.73	1.92	1.65–2.19	2.68	0.001
2.1	Greeting and ID verification	1.68 ± 0.67	4.08 ± 0.83	2.40	2.12–2.68	3.18	0.001
2.2	States purpose of visit	2.01 ± 0.87	3.80 ± 0.87	1.79	1.46–2.12	2.05	0.001
2.3	Explores patient expectations	1.90 ± 0.96	3.46 ± 0.94	1.56	1.20–1.92	1.64	0.001
3	Information gathering	2.65 ± 0.56	3.93 ± 0.64	1.28	1.05–1.51	2.12	0.001
3.1	Uses open questions	2.01 ± 0.99	3.92 ± 0.68	1.91	1.59–2.23	2.24	0.001
3.2	Listens without interrupting	3.95 ± 0.38	4.32 ± 0.37	0.37	0.23–0.51	0.98	0.001
3.3	Summarizes to confirm	1.98 ± 1.11	3.56 ± 1.09	1.58	1.17–1.99	1.43	0.001
4	Patient perspective	1.95 ± 0.56	3.52 ± 1.04	1.57	1.25–1.89	1.87	0.001
4.1	Elicits beliefs about illness	1.75 ± 1.03	3.18 ± 1.13	1.43	1.02–1.84	1.32	0.001
4.2	Explores worries/concerns	2.05 ± 0.99	3.75 ± 1.13	1.70	1.30–2.10	1.60	0.001
4.3	Assesses daily-life impact	2.06 ± 1.05	3.63 ± 1.18	1.57	1.15–1.99	1.40	0.001
5	Information sharing	1.91 ± 0.56	3.57 ± 0.92	1.66	1.37–1.95	2.17	0.001
5.1	Explains diagnosis in plain language	2.28 ± 1.09	3.54 ± 1.05	1.26	0.86–1.66	1.17	0.001
5.2	Uses visual aids/examples	1.91 ± 0.99	3.06 ± 1.33	1.15	0.71–1.59	0.98	0.001
5.3	Checks understanding (teach-back)	1.53 ± 0.92	3.65 ± 1.10	2.12	1.74–2.50	2.09	0.001
6	Plan negotiation	2.03 ± 0.56	3.48 ± 0.97	1.45	1.15–1.75	1.83	0.001
6.1	Discusses therapeutic options	2.23 ± 092	3.61 ± 1.00	1.38	1.02–1.74	1.43	0.001
6.2	Involves patient in decisions	2.11 ± 0.82	3.49 ± 1.13	1.38	1.01–1.75	1.39	0.001
6.3	Negotiates realistic goals	1.75 ± 0.81	3.33 ± 1.11	1.58	1.21–1.95	1.62	0.001
7	Closure	1.84 ± 0.62	3.63 ± 1.05	1.79	1.46–2.12	2.07	0.001
7.1	Provides final summary	1.75 ± 0.77	3.38 ± 1.24	1.63	1.24–2.02	1.57	0.001
7.2	Checks for residual questions	1.70 ± 0.94	3.68 ± 0.99	1.98	1.62–2.34	2.05	0.001
7.3	Closes with empathy and follow-up	2.08 ± 0.92	3.85 ± 1.19	1.77	1.37–2.17	1.66	0.001
8	DM-specific	1.90 ± 0.56	3.96 ± 0.81	2.06	1.80–2.32	2.95	0.001
8.1	Explains lab results (HbA1c)	1.85 ± 0.70	3.93 ± 0.88	2.08	1.78–2.38	2.61	0.001
8.2	Provides initial plan and alleviates anxiety	1.93 ± 0.73	3.99 ± 0.89	2.06	1.75–2.37	2.53	0.001
Total	Overall score (23–115)	49.96 ± 9.72	86.70 ± 17.56	36.74	31.39–42.09	2.58	0.001

**Table 5 ejihpe-15-00152-t005:** Multiple linear regression predicting Δ-score among medical students (n = 30).

Outcome Variable	Cluster A (n = 10)	Cluster B (n = 12)	Cluster C (n = 8)	*p*
Δ-score (total rubric)	58.7 ± 10.2	33.4 ± 11.5	16.2 ± 9.3	<0.001
Relationship (Domain 1)	1.23 ± 0.40	0.68 ± 0.28	0.31 ± 0.17	<0.001
Opening (Domain 2)	1.79 ± 0.52	0.96 ± 0.41	0.37 ± 0.25	<0.001
Information gathering (D3)	1.42 ± 0.61	0.89 ± 0.36	0.34 ± 0.30	<0.001
Patient perspective (D4)	1.51 ± 0.77	0.79 ± 0.45	0.21 ± 0.33	<0.001
Information sharing (D5)	1.76 ± 0.91	1.03 ± 0.59	0.45 ± 0.38	<0.001
Plan negotiation (Domain 6)	1.55 ± 0.87	0.92 ± 0.44	0.33 ± 0.28	<0.001
Closure (Domain 7)	1.62 ± 0.89	0.87 ± 0.53	0.41 ± 0.26	<0.001
DM-specific (Domain 8)	2.04 ± 0.80	1.07 ± 0.62	0.54 ± 0.41	<0.001

## Data Availability

The datasets generated and analyzed during this study—including anonymized rubric scores, survey instruments, AI prompt templates, simulation scripts, and training materials—are publicly available in the Open Science Framework (OSF) repository at https://doi.org/10.17605/OSF.IO/YJF39. Audio recordings and voice files are not publicly accessible due to privacy and ethical restrictions but may be reviewed upon reasonable request and approval by the institutional ethics committee.

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
