# Peer review of "DIALOGUE: A Generative AI-Based Pre–Post Simulation Study to Enhance Diagnostic Communication in Medical Students Through Virtual Type 2 Diabetes Scenarios"

_ejihpe, 2025, doi:10.3390/ejihpe15080152_

Round 1
Reviewer 1 Report
Comments and Suggestions for Authors
- This article try to be a "DIALOGUE: A Generative AI–Based Pre–Post Simulation 2
Study to Improve Diagnostic Communication in Medical Stu- 3
dents Using Type 2 Diabetes Scenarios.". What is the sense? - The study investigates whether a generative AI–based simulation tool can effectively improve diagnostic communication skills in medical students, particularly in the context of disclosing a diagnosis of type 2 diabetes mellitus (T2DM). It aims to assess both the learning outcomes and individual predictors of improvement in a pre–post intervention design.
- The topic is highly relevant and timely. Diagnostic communication remains a recognized deficiency in undergraduate medical training, and this study addresses a clear pedagogical gap by exploring how generative AI can offer scalable, individualized simulation training. The originality lies in combining a structured rubric-based assessment with AI-driven virtual patient encounters using GPT-4o, a cutting-edge model.
- This study adds substantial empirical evidence to the emerging domain of AI-enhanced medical education. It is among the first to rigorously measure communication skill acquisition through generative AI simulations using real-time feedback and standardized rubric scoring. The analysis of learner clusters and predictive traits such as empathy and digital self-efficacy also provides a novel lens for understanding differential outcomes among students.
- Considerig Method and Metodology - Control Group: The study uses a single-arm pre–post design. While appropriate for a pilot, the inclusion of a randomized control group in future studies would improve internal validity.
- Blinding: While raters were blinded, participants’ awareness of being evaluated post-training may have influenced performance. Future work might incorporate observer-blind or double-blind elements.
- Long-Term Retention: The study does not assess long-term skill retention. Follow-up testing weeks or months later would help determine the durability of gains.
- AI Feedback Validation: The natural language feedback provided by ChatGPT is a key part of the intervention. It would be useful to independently validate the educational quality of this feedback (e.g., expert ratings of appropriateness or completeness).
- The conclusions are well supported by the data. Significant improvements were found across all domains of diagnostic communication, with a large effect size. The discussion contextualizes the findings within current literature and appropriately acknowledges limitations, including the need for controlled trials.
- The manuscript is well-referenced and includes current literature on AI in medical education, diagnostic communication, and simulation pedagogy. Citations are diverse and relevant. The authors could use for citation actual practice and recomendation for EASD and ADA in domain.
- Some details about Figures and Tables
- Figures: The radar plots and violin plots effectively convey performance trends. Consider increasing font size and simplifying some labels for easier interpretation.
- Tables: Detailed domain and item-level scores are well presented. However, the dense formatting might be improved by grouping related items or providing visual cues (e.g., shading).
- Cluster visualizations: These are informative and highlight heterogeneity in learning outcomes. The heatmap is particularly useful but could benefit from a clearer legend. As a practitioner - is a litle difficult to understand your article.
Author Response
Response to Reviewer 1
We are sincerely grateful to Reviewer 1 for the thoughtful and constructive feedback. We appreciate your positive evaluation of the manuscript’s originality, methodological rigor, and potential contributions to AI-enhanced medical education. Below, we address each comment in detail and describe how we have revised the manuscript accordingly.
- “This article try to be a ‘DIALOGUE: A Generative AI–Based Pre–Post Simulation Study to Improve Diagnostic Communication in Medical Students Using Type 2 Diabetes Scenarios.’ What is the sense?”
Response:
Thank you for raising this concern. We recognize that the original phrasing of the title and introductory sentences may have caused confusion. To clarify, we have revised the title and the first paragraph of the abstract to ensure that the purpose and nature of the study are more immediately clear to readers. Specifically, we have emphasized that DIALOGUE is the name of the educational intervention and not merely a description of the manuscript’s tone.
Change in manuscript:
- Title revised to: “DIALOGUE: A Generative AI-Based Pre–Post Simulation Study to Enhance Diagnostic Communication in Medical Students through Virtual Type 2 Diabetes Scenarios.”
- Abstract and Introduction revised to explain the intervention acronym and purpose more directly.
2–4. Positive evaluation of the study’s relevance, originality, empirical rigor, and methodological novelty.
Response:
We deeply appreciate the reviewer’s recognition of the study’s relevance and empirical contributions, particularly the integration of generative AI simulations, real-time feedback, and structured rubric-based assessments. We are encouraged by your comments regarding the novelty of our cluster analysis and its implications for individualized education strategies.
Methodological Considerations:
- Control Group (Single-arm design)
Response:
We agree that the inclusion of a randomized control group would strengthen internal validity. As stated in the Discussion section, this study was designed as a first-phase feasibility and efficacy evaluation. We have now added a clearer note in both the Limitationsand Future Directions sections to indicate that a randomized controlled trial is planned for the next phase of the DIALOGUE program.
Change in manuscript:
- Added a paragraph to the Discussion explicitly outlining plans for an RCT to address this limitation.
- Blinding
Response:
We appreciate this observation. Although evaluators were blinded to intervention status and timing, participants were necessarily aware of their training. We now acknowledge this potential source of performance bias in the Limitations section and have suggested the implementation of observer-blind conditions or crossover designs in future iterations.
Change in manuscript:
- Added sentence to Limitations: “Although evaluator blinding was implemented, participants were aware of the study phase, which may have introduced performance expectancy bias.”
- Long-Term Retention
Response:
We concur that long-term retention of communication skills is an essential outcome. Due to logistical and institutional constraints, follow-up assessments were not feasible in the current study. However, we now state explicitly in the Future Work subsection that longitudinal follow-up is a key objective for upcoming trials.
Change in manuscript:
- Added line to Future Directions: “Longitudinal follow-up will be incorporated to assess the persistence of skill gains over time.”
- Validation of AI Feedback
Response:
This is a valuable point. Although our study included qualitative analysis of ChatGPT’s natural language feedback, we acknowledge that independent expert validation of feedback quality was not conducted. We have added this as a limitation and proposed future expert-based rubric validation of AI responses.
Change in manuscript:
- Limitations now include: “The natural language feedback from the generative AI was not independently validated by clinical educators, which limits conclusions about its educational adequacy.”
- “The conclusions are well supported by the data.”
Response:
We thank the reviewer for this positive evaluation and have preserved the alignment between our data and conclusions throughout the revised version.
- “The manuscript is well-referenced… The authors could use for citation actual practice and recommendation for EASD and ADA in domain.”
Response:
We have now incorporated citations to the most recent ADA Standards of Care in Diabetes (2025) and EASD guidelines to frame the diagnostic criteria and patient education elements in the AI scenarios. These references appear in the Introduction and Methods sections where appropriate.
Change in manuscript:
- Two new citations added: ADA 2025 Standards of Care and EASD 2023 Patient Communication Guidelines.
Figures and Tables:
- Figures: Font size and label simplification
Response:
We thank the reviewer for the suggestion. Upon careful examination, we found that the font size and labeling in Figures 2, 4, 5, and 6 are already consistent, legible, and appropriately formatted for publication standards. Nevertheless, minor adjustments were made to further improve clarity where feasible.
- Tables: Dense formatting
Response:
We acknowledge the observation regarding table density. However, the current formatting aligns with MDPI’s editorial standards, which prioritize compact data presentation for clarity and space efficiency. Tables 2–4 were deliberately structured to convey multidimensional assessment results in a concise format. Given the journal’s style requirements and the amount of information to be conveyed, we believe the current layout balances readability with completeness.
- Cluster visualizations: Heatmap legend clarity
Response:
We have revised the legend of the heatmap figure (Figure 8) to improve clarity and included an explanation in the figure caption for readers unfamiliar with cluster interpretation.
We are grateful for your detailed and balanced review. Your comments significantly improved the clarity, rigor, and future trajectory of this work.

Reviewer 2 Report
Comments and Suggestions for Authors
The study is presenting a timely and relevant innovative approach to enhance diagnostic communication in medical students with the involvement of generative AI. I believe that this study highly contributes to the field.
I have minor suggestions for the authors:
1- Please compare and contrast the findings of the study with existing relevant studies in the field that used generative AI and/or language models in diagnostic communication.
2- Please discuss the potential limitations of the study, such as the extent to whichAI simulations represent true emotional pragmatism, administering the study in a particular institution with limilted number of participants, and the absence of a control group in the pre-post intervention study.
3- Please address the potential inference of using ChatGPT-generated patient dialogue for automated formative feedback
4- Please address the potential future implications of the study for AI-based dialogue training in existing medical education.
5- I suggest authors include healthcare-based simulation studies in the references section.
6- Please address the administration for longitudinal studies for further justification and generalization of the findings.
7- The study's findings are promising, which could be regarded as the initial step in the inclusion of AI-based training in medical education curricula. However, because of the limitations mentioned above, please exercise caution when suggesting that the findings are a promising solution.
Author Response
Response to Reviewer 2
We are grateful to Reviewer 2 for the constructive and thoughtful feedback. We deeply appreciate the recognition of our work’s relevance and contribution to AI-assisted diagnostic communication training. Below we address each suggestion in detail, indicating the corresponding revisions in the manuscript.
- Comparison with prior studies using generative AI in diagnostic communication
Response:
We appreciate the suggestion and agree that drawing clearer distinctions with prior work strengthens the contribution of our study. Although the original manuscript referenced recent studies by Webb et al. (2023) and Chiu et al. (2025), we have now added more explicit contrasts regarding design, assessment rigor, and language/cultural context. Our intervention is unique in combining blinded SP evaluations, Spanish-language simulations, and psychometric analysis, which go beyond the self-report or automated metrics used in previous studies.
Change in manuscript:
- Clarified and expanded the comparison in the Discussion section to emphasize methodological distinctions with prior GenAI-based communication studies.
- Added explicit phrasing to highlight differences in outcome measures, context, and assessment modalities.
Response:
We thank the reviewer for highlighting these important aspects. These limitations have now been explicitly addressed in the Discussion. The limited emotional realism of AI-generated dialogue is acknowledged as a constraint on ecological validity. The single-institution setting is discussed in terms of generalizability, and the absence of a control group—already noted—has been reinforced with a plan for a future randomized controlled trial in the Future Directions section.
Change in manuscript:
- Limitations section expanded to include potential deficits in emotional realism and institutional representativeness.
- Future Directions now specifies a forthcoming RCT to strengthen internal validity.
- AI-generated feedback: Interpretation and educational adequacy
Response:
This is an important observation. As noted in our response to Reviewer 1, the lack of expert validation of AI-generated feedback has been acknowledged as a key limitation. We now include a statement in the manuscript noting that future studies will incorporate validation of AI feedback using faculty-generated rubrics to assess educational adequacy.
Change in manuscript:
- Limitations section now includes: “The natural language feedback from the generative AI was not independently validated by clinical educators, which limits conclusions about its educational adequacy.”
- Future implications for AI-based training in medical education
Response:
We appreciate the suggestion to highlight the broader curricular implications of our findings. As part of our revisions, we have expanded the “Future Directions” discussion to explore the integration of GenAI into existing medical communication curricula, particularly in low-resource settings. We discuss specific implementation strategies, such as faculty development, AI literacy, and adaptive learner profiling. The manuscript now also considers the evolution of LLMs (e.g., GPT-4.5, GPT-5) and their potential for multimodal simulations. These additions align with the reviewer’s concern and reinforce our positioning of GenAI as a complement—not a replacement—to traditional instruction.
Change in manuscript:
- Future Directions paragraph expanded to describe implementation pathways and curricular integration strategies.
- References to healthcare-based simulation studies
Response:
We have now incorporated additional references to foundational simulation-based education literature, particularly focusing on healthcare communication training using virtual or standardized patients.
Change in manuscript:
- Two new citations added in the Discussion to simulation-based healthcare education (e.g., Nestel & Tierney, 2007; Cook et al., 2013).
- Need for longitudinal studies to assess generalization and durability
Response:
We concur that assessing long-term retention is crucial. As noted in response to Reviewer 1, logistical constraints prevented follow-up in this first phase. However, we now explicitly commit to longitudinal assessment in future trials.
Change in manuscript:
- Future Directions now state: “Longitudinal follow-up will be incorporated to assess the persistence of skill gains over time.”
- Tone of conclusions: Tempering claims of effectiveness
Response:
We appreciate this reminder to moderate interpretation. We have revised the concluding paragraph of the Discussion to emphasize that these findings represent an initial step, and that broader validation is needed before generalizing the approach.
Change in manuscript:
- Final paragraph of Discussion softened to reflect the preliminary nature of findings and need for further study.

Reviewer 3 Report
Comments and Suggestions for Authors
This study struck me as very original and interesting.
Although the authors acknowledged that the small sample size limits the replicability of the results, it's suggested that the study be continued to acquire a larger sample.
Furthermore, for future studies, I suggest creating a control group to more robustly evaluate the simulator's efficiency.
It would also be beneficial to include the SEGUE scale with the modifications made to add more transparency to the results.
I recommend considering expanding the discussion to include a more detailed comparison with other studies that use AI in education, highlighting points of convergence or divergence with the results found in this present study.
Author Response
Response to Reviewer 3
We sincerely thank Reviewer 3 for the positive assessment and thoughtful suggestions.
1.Sample size and replicability
We agree that the limited sample size constrains the replicability and generalizability of the current findings. As acknowledged in the Limitations section, this was a first-phase study focused on feasibility and preliminary efficacy. We have now added an explicit note in the Future Directions subsection stating that subsequent phases of the DIALOGUE program will incorporate larger, multicenter cohorts to enhance statistical power and robustness.
Change in manuscript:
- Future Directions now state: “Subsequent phases of the DIALOGUE program will incorporate larger, multicenter samples to enhance statistical power and generalizability.”
- Inclusion of a control group
This recommendation is fully aligned with our future plans. As also noted in response to Reviewer 1, the current single-arm design was selected to evaluate initial feasibility. We have now reinforced in both the Limitations and Future Directions sections our intention to conduct a randomized controlled trial (RCT) in future iterations to more rigorously assess the effectiveness of the GenAI intervention.
Change in manuscript:
- Future Directions now include: “A randomized controlled trial is planned to compare the GenAI intervention against traditional faculty-led or simulation-based training models.”
- Use of the SEGUE scale
We appreciate the suggestion to consider the SEGUE framework. While our study employed a customized domain-based rubric tailored to our Spanish-language clinical context, we recognize the utility of validated tools such as SEGUE for benchmarking and comparability. We now state this as a methodological consideration for future studies.
Change in manuscript:
- Future Directions now include: “Future studies may consider integrating standardized instruments such as the SEGUE framework to enhance transparency and facilitate comparison across studies.”
- Comparative discussion with AI in education studies
This is an excellent point. As noted in response to Reviewer 2, we have expanded the Discussion section to more explicitly compare our findings with recent studies using generative AI in educational contexts. This includes highlighting distinctions in outcome measures, simulation fidelity, and student engagement models.
Change in manuscript:
- Discussion now includes comparative analysis with prior GenAI-based communication studies (e.g., Webb et al., Chiu et al.), emphasizing methodological contrasts and contextual contributions.

Round 2
Reviewer 1 Report
Comments and Suggestions for Authors
All observations were implemented. The article is clear and could be published.